Herbivore camping reshapes the taxonomy, function and network of pasture soil microbial communities

Wang Puchang 1
Ding Leilei peterding2007gy@163.com 2
Li Fuxiang 3
Liao Jiafa 3
Wang Mengya 4
1 School of Life Sciences, Guizhou Normal University , Guiyang , Guizhou , The People’s Republic of China
2 Guizhou Institution of Prataculture, Guizhou Academy of Agricultural Sciences , Guiyang , Guizhou , The People’s Republic of China
3 Guizhou Weining plateau Grassland Experimental Station , Weining , Guizhou , The People’s Republic of China
4 College of Animal Science, Guizhou University , Guiyang , Guizhou , The People’s Republic of China
Tomar Mahendra
Electronic publication date: 2022 Nov 9
Publication date: 2022
Volume: 10
Electronic Location ID: e14314
Received 2022 Jul 14; Accepted 2022 Oct 7
Copyright: ©2022 Wang et al.
Copyright year: 2022
Copyright holder: Wang et al.
License: This is an open access article distributed under the terms of the Creative Commons Attribution License, which permits unrestricted use, distribution, reproduction and adaptation in any medium and for any purpose provided that it is properly attributed. For attribution, the original author(s), title, publication source (PeerJ) and either DOI or URL of the article must be cited.
License URL: https://creativecommons.org/licenses/by/4.0/

Keywords: Fungi, Bacteria, Grassland, Network robustness, Niche breadth, Niche overlap, Camping, Soil, Community structure

Funding: National Nature Science Foundation of China 31960341 Technology Foundation of China Qiankehejichu-ZK[2021]yiban157 Guizhou Province Science Qiannongkeyuanguojihoubuzhu project (2021)03 This research was supported by the National Nature Science Foundation of China (31960341), the Guizhou Province Science and Technology Foundation of China (Qiankehejichu-ZK[2021]yiban157) and the Qiannongkeyuanguojihoubuzhu project (Qiannongkeyuanguojihoubuzhu(2021)03). The funders had no role in study design, data collection and analysis, decision to publish, or preparation of the manuscript.

==============================
Although the effects of herbivore camping on soil physicochemical properties have been studied, whether the effects alter the soil microbial communities (e.g., composition, functions, taxonomic and functional diversities, network) remain unknown, especially below the surface. Here, using paired subsoil samples from half month-camping and non-camping, we showed for the first time that camping significantly changed the relative abundance of 21 bacterial phylotypes and five fungal phylotypes. Specifically, we observed significant increases in the relative abundance of putative chitinase and terpenes vanillin-decomposition genes, nitrite reduction function (nirB, nasA), decreases in the relative abundance of putative carbon fixation genes (ackA, PGK, and Pak), starch-decomposition gene (dexB), gene coding nitrogenase (anfG), and tetracycline resistance gene (tetB) for bacterial communities, and significant decreases in the relative abundance of animal endosymbiont and increases in the relative abundance of litter saprotroph and endophyte for fungal communities. However, camping did not significantly impact the taxonomic and functional diversity. The niche restriction was the main driving force of bacterial and fungal community assembly. Compared to no camping, camping increased the stability of bacterial networks but decreased the stability of fungal networks. Camping exerted a positive effect on the network by compressing the niche width and reduced the change in the network by reducing the niche overlap. Our results suggest that camping restructures the soil microbial composition, function, and network, and provides a novel insight into the effect of animal camping on soil microbial communities in grassland.

Introduction

Camping can involve many fields, including military, ecotourism recreation, wildlife habitat behavior and grassland management. In the field of grassland management, herbivore camping is considered an important part of the technology of improving and restoring degraded grassland by herbivore night penning, which removes undesired vegetation, and changes the nutrient content and structure of soils through herbivore excretion and trampling during night penning, as well as subsequent sowing (Jiang, Wa & Liu, 1996; Jiang et al., 1999; Zhang, Jiang & Ren, 2001; Yuan et al., 2012). Indeed, herbivore camping is a successful comprehensive technology (Zhang, Jiang & Ren, 2001) that can improve natural grasslands and rebuild artificial grassland at a very low cost (Zhang, 2002; Yuan et al., 2012). This type of camping has been used in grassland and pasture management in New Zealand, North Korea, Inner Mongolia, Guizhou, and Yunnan in China.

Previous in-depth explorations have advanced the understanding of the effects of camping on vegetation and soil (Jiang, Wa & Liu, 1996; Jeffrey & David, 1996; Haynes & Williams, 1999; Zhang, Jiang & Ren, 2001; Zhang, 2002; Niu et al., 2009), and suggested that camping can facilitate grass, but suppress shrubs and weeds (Zhang, Jiang & Ren, 2001; Zhang, 2002; Yuan et al., 2012). Additionally, camping is known to improve soil fertility and pH (Jiang, Wa & Liu, 1996; Haynes & Williams, 1999; Zhang, Jiang & Ren, 2001; Zhang, 2002; Niu et al., 2009; Yuan et al., 2012) with the magnitude of impact found to vary with camping intensity and topographic position (Jeffrey & David, 1996). Zhang et al. (1999a) extended the effect of camping on plants to plant physiology, and found that sheep manure and urine change the plant water potential and the cell membrane permeability, although the feces and urine alone does not have the same effect. Moreover, Iyyemperumal, Israel & Shi (2007) and Haynes & Williams (1999) used substance transformation rates and enzyme activities to determine the microbial activity, which they suggested could be altered by camping. However, changes in microbial activity do not inform changes in the structure, function, and network of bacterial and fungal communities. Although the effects of camping on plants, soil physicochemical properties, and soil enzyme activities have been well studied, whether and how camping affects the structure, function, and network of soil microbial communities has not yet been explored.

Inevitably, the effects of camping on plants and soil have the combined effects of browsing, trampling, and the input of feces and urine (Haynes & Williams, 1999; Zhang, Jiang & Ren, 2001; Iyyemperumal, Israel & Shi, 2007; Niu et al., 2009). Jiang et al. (1999), Zhang (2002), and Hou et al. (2004) reported that browsing and trampling reduce plants and litter, and change soil physical properties, while Liu et al. (2015) found that trampling increases the abundance of soil bacteria, fungi, and arbuscular mycorrhizal fungi. An increasing number of studies have also shown that inputs of dung and urine are crucial pathways by which herbivores enhance soil fertility (Yu, Nan & Hou, 2008), change plant nutrients (Williams et al., 1999; Yu, Nan & Hou, 2008; Li et al., 2021a) and improve yield (Shi et al., 2021; Asmita et al., 2021), increase soil microbial activity and biomass (Williams et al., 1999; Rooney et al., 2006; Liu et al., 2020), and substantially alter the soil microbial community structure (Rooney et al., 2006; Rooney & Clipson, 2008; Liu et al., 2020; Li et al., 2021a; Liu et al., 2021). Despite this evidence to support salubrious effects of excrements in soil and plants (Rayne & Aula, 2020), as the microbial community composition in dung is different and less diverse than that in soil (Andrea et al., 2021), animal dung input into soils can introduce dung-derived microbes (Macedo et al., 2021) and the propagation and proliferation of antibiotic resistance genes into soil, which likely constitute a serious threat to ecosystem and human health (Zhang et al., 2021). Therefore, it is important to understand the effects of camping on soil microorganisms and antibiotic resistance genes. Although the effects of browsing, trampling, and inputs of dung and urine on soil have been well studied individually, as complex interactions exist between these components (Li et al., 2021a), their individual effects are difficult to extrapolate to the overall effect of camping; thus, it remains unknown how camping alters soil bacterial and fungal communities in the real world.

Therefore, the aims of this study were to decipher whether herbivore camping changes the taxonomic, functional, and network patterns of soil bacterial and fungal communities. We hypothesized that camping reshapes the structure, function, and interactive network attributes of soil bacterial and fungal communities. Specifically, this study answers the following questions for the first time: (1) Does the camping enhance bacterial and fungal diversity? (2) What bacteria and fungi are enriched and inhibited by camping and what are their functional implications? and (3) Does camping improve the bacterial and fungal interaction network, and if so, how? This study provides novel insights into the effects of camping on pasture ecosystems that experience herbivore camping, common to grasslands worldwide, and supplies updated knowledge for understanding this technology.

Material and Methods

Sampling sites

The study sampling sites were located in Weining county (26°52′N, 104°17′E; ca. 2,440 m above sea level), Guizhou Province, SW China (Zhang, Jiang & Ren, 2001). This area experiences a subtropical and warm temperate monsoon climate, with a mean annual rainfall of 926 mm and temperature of 11.6 °C, a frost-free period of 180 d, and average sunshine hours of 1800 (Wu & Shen, 2020). Yellow brunisolic soil covers this area (Zhang, Jiang & Ren, 2001). Basic information on the nutrient content of grassland soil is listed in the Supplementary Materials.

Sheep for camping

To understand the effects of animal camping in the real world, Wumeng semi-fine wool sheep were used for camping, since these accounts for 62% of the sheep in the stock of Guizhou (Wu & Shen, 2020) and they are of economic significance in the Wumeng mountain area (Wu, Song & Shen, 2020) of Guizhou (Wu & Shen, 2020). However, sheep camping is a spontaneous and important part of grassland management for local herders.

Design and soil sampling

The sheep camping (ca. half sheep/night/m2) was conducted spontaneously for 15 nights (Jiang, Wa & Liu, 1996) by local herdsmen in October 2018. On the 16th day, we only collected subsurface soil (10–20 cm) samples from three paired sites ((camping sites and corresponding no camping sites) ×3), where the topographies, plants, and soil types are nearly identical. Three samples were collected randomly from each site and mixed into a composite sample; this sampling method reduced to the maximum extent the impacts of non-design source variation (e.g., background environmental variation). To avoid contamination, sampling at each site was conducted using 75%-alcohol sterilized and separate cut rings with hands wearing separate medical sterile gloves. Soil samples were kept in separate sterile plastic bags on dry ice during transportation, and stored at −80 °C until DNA extraction.

DNA extraction, sequencing of amplicons, processing of sequence data, and bioinformatics analysis

To determine whether camping affected soil microbial communities, soil DNA was isolated by a PowerSoil DNA Isolation Kit, and its quality and quantity were checked by electrophoresis. PCR amplification was performed using an Applied Biosystems Gene Amp PCR System 9700. The V4 region of the bacterial 16S rRNA was PCR-amplified using the primers 806R and 515F, and the fungal ITS2 region was amplified using the primers ITS3_KYO2 and ITS4. The amplified products were extracted using Qubit 2.0 (Thermo Fisher Scientific, Waltham, MA, USA). Illumina sequencing was conducted using an Illumina HiSeq platform with PE250 mode (Illumina, Inc, San Diego, CA, USA). The original sequencing data were spliced and filtered to obtain high-quality sequences. Usearch (http://drive5.com/uparse/) was performed to obtain OTU matrices at a cut-off of 97%. To decipher whether camping changes the taxonomic patterns of soil microbial communities, taxonomy was assigned using Uclust (https://drive5.com/usearch/manual/uclust_algo.html) based on the Silva database (https://www.arb-silva.de/) for the bacterial community, or the Unite database (https://unite.ut.ee/) for the fungal community (Ding et al., 2020). To normalize the disparity in the sequence number (Ding et al., 2020), bacterial communities were rarefied to 10,432 sequences and fungal communities were rarefied to 9,236 sequences for each sample. To understand the effects of camping on soil microbial biodiversity, Good’s coverage, observed OTUs, Chao1, ACE, InvSimpson (Inverse Simpson), Fisher’s diversity and Faith’s phylogenetic (PD) diversity were calculated. To decipher whether camping changes the functional patterns of soil microbial communities, the functional profiles and diversity were calculated. The methods used to retrieve the functional profiles and diversity of bacterial and fungal communities followed the methods used in our previous study (Ding & Wang, 2021) with minor modifications. Briefly, the soil bacterial functional profile was predicted using the “Tax4Fun” package (http://tax4fun.gobics.de/Tax4Fun/Tax4Fun_0.3.1.zip) in R version 3.0 (https://cloud.r-project.org/), and FAPROTAX version 1.2.4 (https://pages.uoregon.edu/slouca/LoucaLab/archive/FAPROTAX/lib/php/index.php?section=Download) in Python version 3.7.4 (https://www.python.org/downloads/release/python-374/). The fungal functional profiles were predicted using FUNGuild (http://www.funguild.org/) in Python version 3.7.4. The richness, Shannon, Pielou, and InvSimpson of functions were calculated using the “vegan” package (https://cloud.r-project.org/) in R version 3.6.

Statistical analysis

To determine the difference in soil microbial attributes between the camping and no camping groups, ANOVA and t test (Wilson, 1927) were used when data met the requirements of normality and homoscedasticity; otherwise, the Kruskal–Wallis test and Wilcoxon test (Myles & Douglas, 1973) were performed (Ding et al., 2020). Shapiro–Wilk test and Levene’s test (Royston, 1982) were applied to determine the normality and homoscedasticity, respectively (Ding & Wang, 2021).

To determine the important phylotypes with significant differences between groups, the random forest analysis (“randomForest” package) (Breiman, 2001) and Kruskal–Wallis test (“stats” package) (Myles & Douglas, 1973) were used in R (https://www.r-project.org/). Welch’s t test with two sides was used in STAMP (https://beikolab.cs.dal.ca/software/STAMP) to understand the functional differences in soil microbial communities between groups. Permutational multivariate analysis of variance (PERMANOVA) was used to understand differences in the soil bacterial or fungal community composition between groups at the overall community level. The dispersal–niche continuum index was applied to quantify the relative role of the niche or dispersal process that shaped community assembly (Vilmi et al., 2020). The niche breadth and overlap were calculated using the “spaa” package (https://github.com/helixcn/spaa).

Network analysis is a robust and effective method to quantify microbial interactions (Kong et al., 2019; Zhou et al., 2020), and was therefore used to determine the effects of camping on microbiome interactions. To minimize the impact of data sparsity on the network and strengthen the reliability of the network, only bacterial or fungal OTUs appeared in ≥ 2 samples (Banerjee et al., 2019) and had a total relative abundance ≥ 0.1% in all samples were included in the network analysis (Zhang et al., 2018); this resulted in 881 bacterial and 183 fungal OTUs. Microbial interaction networks were constructed using the “WGCNA” (Langfelder & Horvath, 2008) and “igraph” (https://igraph.org/r/) packages. The Pearson correlation coefficient was calculated for each pair of OTUs, and the Benjamini and Hochberg corrected p value (Benjamini & Hochberg, 1995) was reported (Fan et al., 2020). Random matrix theory was used to determine the optimal threshold of correlations in constructing microbial interaction networks, as outlined previously (Zhou et al., 2020). A threshold of strong Pearson’s r  > 0.8 and p < 0.001 was set based on the following three facts: (1) The microbial phylotypes that show strong correlations with each other are more likely to interact with each other (Fan et al., 2020); (2) The optimal threshold from random matrix theory and priori knowledge (Fan et al., 2020; Yuan et al., 2021) is approximately 0.8; (3) The network properties were must be compared under the same conditions (Zhou et al., 2020).

To understand the effects of camping on soil microbial interaction network attributes, several tests were performed as follows: First, a two-sided Kolmogorov–Smirnov test (William, 1971) was used by performing the “ks.test” function to discriminate the cumulative distribution of 10,000 bootstrapping node properties of the camping and no camping networks; the null hypothesis was that the properties under the camping and no camping have same distribution patterns (Banerjee et al., 2019). Second, a Kolmogorov–Smirnov test and Kruskal-Wallis test were employed to determine the difference in robustness (average degree and natural connectivity) between the camping and no camping networks after 50% of the nodes were stochastically removed (Banerjee et al., 2019; Yuan et al., 2021). The network robustness, defined as the declines in microbial average degree (and natural connectivity) with the increasing proportion of removing nodes (and edges), was also tested (Pan et al., 2021; Shi et al., 2021). The linear regression model was used to describe these relationships (Chambers, 1992) and to test the effect of niche width and niche overlap on the network property [degree and difference between degrees (Δ degree)].

To decipher whether camping changes the network of soil microbial communities by changing the niche width and overlap, the Bayesian structural equation model (Bürkner, 2017) was implemented to examine potential pathways that can account for how camping alters the network property. A pathway was considered acceptable if the 95%CI (confidence interval) of the coefficient of the pathway did not contain 0 (Bürkner, 2017).

Results

Diversity and composition of soil microbial communities under camping and no camping

The normality and homoscedasticity tests showed that the observed OTUs, Chao1, ACE, InvSimpson, Fisher, and PD of the soil microbial communities followed the premises of normality and homogeneity (Shapiro–Wilk’s test and Levene’s test, p = 0.0610–0.9661, Table S1). As expected, the ANOVA and t test showed that all diversity indexes in the camping group were not significantly distinct from the no camping group (p = 0.308–0.901, Table S2). Unexpectedly, based on Bray–Curtis (R2 = 0.224, p = 0.20 for bacteria; R2 = 0.192, p = 0.50 for fungus), jaccard (R2 = 0.220, p = 0.20 for bacteria; R2 = 0.225, p = 0.30 for fungus), unweighted unifrac (R2 = 0.214, p = 0.40 for bacteria; R2 = 0.204, p = 0.50 for fungus) and weighted unifrac (R2 = 0.169, p = 0.60 for bacteria; R2 = 0.222, p = 0.50 for fungus) distances, the PERMANOVA test with 9,999 permutations showed no significant difference in the soil bacterial or fungal community composition between groups at the overall community level (Fig. S1). However, Venn plots showed that 385 (1.52% of total sequence number) and 390 (1.46% of total sequence number) bacterial OTUs and 259 (2.96% of total sequence number) and 192 (1.13% of total sequence number) fungal OTUs were unique to camping and no camping, respectively (Fig. S2). Furthermore, random forest analysis showed that these two groups could be clearly predicted and significantly separated by 21 bacterial phylotypes and five fungal phylotypes (Kruskal–Wallis test, p = 0.0369–0.0495, Fig. 1). Camping significantly increased the abundance of Chelatococcus, Luteimonas, Dyadobacter, Ruminococcaceae UCG-005, Aquamicrobium, Cohnella, Limnothrix, Pseudanabaena PCC-7429, Family XIII AD3011 group, Christensenellaceae R-7 group, Prevotellaceae UCG-004, Olivibacter, OLB13, and Rummeliibacillus, but significantly reduced the abundance of Ruminococcus 1, LD29, Rudaea, RB41, FukuN18 freshwater group, Cellvibrio, and Clostridium sensu stricto 18 in bacterial communities. However, the camping significantly increased the abundance of Podospora, Rhexocercosporidium, and Symbiotaphrina, but significantly decreased the abundance of Rhizomucor and Candida in fungal communities (Kruskal–Wallis test p = 0.0369–0.0495), compared to no camping.

Figure 1 Random Forest analysis (left) and Kruskal–Wallis test (right) of bacterial (A) and fungal communities (B).

Mean decrease Gini: mean decrease in Gini index. The larger the value, the more important the phylotypes is in distinguishing groups. *, p < 0.05.

Functional profiles and diversity of soil microbial communities under camping and no camping

Contrary to the intuitive, as much as 83 function group levels related to C (Figs. S3–S16, and S57–S59), N (Figs. S18–S31, and S60–S65), P (Figs. S66–S70), hydrogen (Figa. S32), sulfur (Figs. S33–S37), iron (Fig. S38), manganese (Fig. S39) cycles, plastic degradation (Fig. S17), fermentation (Fig. S40), plant pathogens (Fig. S41), animal parasites or symbionts (Fig. S42), predatory and/or parasites (Figs. S43–S45), mammal gut bacteria (Fig. S46), bacterial nutrition types (Figs. S47–S54), and antibiotic resistance genes (Figs. S71–S82) were tested using Wilcoxon rank sum test. The results of these tests suggested no significant differences in the diversity indices (richness, Shannon, Pielou, InvSimpson) of these bacterial functions between camping and no camping (Figs. S3–S82). However, no significant differences were detected in the diversity indices (richness, Shannon, Pielou, InvSimpson) of the growth form, guild, trait, and trophic mode of fungus (Figs. S83–S86). However, bacterial functional analysis showed that compared to no camping, camping significantly increased the relative abundance of C decomposition genes (chitinase and Terpenes vanillin) and nitrite reduction function (nirB, nasA), and significantly decreased the relative abundance of C fixation genes (ackA, PGK, Pak), C decomposition gene (dexB), gene coding nitrogenase (anfG), and tetracycline resistance gene (tetB) (Welch’s t test with two sides p = 8.95e−4–0.046, Figs. 2A–2D). Moreover, camping significantly increased the relative abundance of dark thiosulfate oxidation and cellulolysis, but significantly decreased the relative abundance of xylanolysis (Welch’s t test with two sides p = 0.032–0.048, Fig. 2E). Camping significantly decreased the relative abundance of animal parasites or symbionts (Welch’s t test with two sides p = 0.032), while fungal functional analysis showed that camping significantly decreased the relative abundance of animal endosymbiont (Welch’s t test with two sides p = 0.030). Additionally, camping significantly increased the relative abundance of litter saprotroph and endophytes (Welch’s t test with two sides p = 0.013, 0.018, Fig. 2F).

Figure 2 Functional differences of soil microbial communities (A–E: bacteria, F: fungi) between camping and no camping. C, carbon-related function; N, nitrogen-related function; AR, antibiotic resistance gene.

Soil microbial networks under camping and no camping

To determine the effects of camping on microbiome associations, bacterial and fungal networks were established under camping and no camping treatments. The results revealed distinct association patterns (Fig. 3). Compared to the no camping soils, although camping decreased the number of nodes (by 3%, 19%), number of clusters (9%, 10%), centralization degree (4%, 9%), and negative edge proportion (4%, 39%) of bacterial and fungal networks, and elevated the positive edges proportion (3%, 27%) and modularity (0.1%, 37%) of the bacterial and fungal networks. Camping decreased the central eigen (by 1%) and vulnerability (11%) of the bacterial network, but increased those (15%, 44%) of the fungal network. Camping also increased the number of edges (by 5%), connectance (11%), average degree (8%), number of positive edges (8%), number of negative edges (1%), and natural connectivity (0.01%) of the bacterial network, but increased those (49%, 21%, 36%, 35%, 69%, 38%) of the fungal network (Table S3). Kolmogorov–Smirnov test indicated that the node degree, closeness, transitivity, and eigenvector centrality under camping were statistically distinct from those under no camping (p = 2.2e−16–4.62e−8, Table 1). We assessed the difference in network stability (average degree and network connectivity) between the camping and no camping treatments by network bootstrapping after 50% of the nodes were randomly removed. Kolmogorov–Smirnov test indicated that camping significantly changed the network stability (average degree and network connectivity, p = 2.2e−16, 2.2e−16), while Kruskal test revealed that the average degree and network connectivity of the bacterial network were 1.9–3.9-fold those of the fungal network, regardless of the presence or absence of camping (Table 2). Interestingly, camping significantly increased the average degree and network connectivity of the bacterial network by 5% and 1% (p = 2.2e−16, 2.2e−16), respectively. Nevertheless, camping significantly decreased that of the fungal network by 50% and 40% (p = 2.2e−16, 2.2e−16), respectively. We next performed robustness analysis of the networks based on removing a proportion of nodes and edges. The results showed that the average degree and network connectivity of the bacterial network were higher than those of the fungal network, regardless of the presence or absence of camping. Compared to no camping, camping increased the average degree and network connectivity of the bacterial network, but decreased those of fungal network, irrespective of the removal of nodes and edges (Fig. 4).

Figure 3 Co-occurrence networks for microbial communities between camping and no camping.

A node suggests an OUT, its colour and size are proportional to its degree; a link represents the significant Pearson correlations with r > 0.8 and the Benjamini and Hochberg corrected p < 0.001. A red link indicates a positive relationship, but a blue link indicates a negative relationship.

Table 1 Results of the Kolmogorov–Smirnov test comparing bootstrapped node attributes of networks under No camping and Camping.

For each network, node attributes were computed by bootstrapping 100,000 times. Kolmogorov–Smirnov test compares the cumulative distribution of two properties where the null hypothesis is that the properties have same distribution patterns.

Comparison	Degree	Closeness	Transitivity	Eigenvector centrality	
No camping B vs.
Camping B	0.1665****	0.5276****	0.0133****	0.2984****	
No camping F vs.
Camping F	0.4941****	0.5066****	0.0595****	0.2462****	
No camping B vs.
No camping F	0.4693****	1****	0.0967****	0.3248****	
Camping B vs.
Camping F	0.4913****	1****	0.1430****	0.2604****	
Notes.

The values in each cell represents the maximum difference in the absolute cumulative distribution function.

**** indicate statistical significance at p <0.0001, respectively.

B bacteria

F fungi

Table 2 Results of the Kolmogorov–Smirnov test and Kruskal–Wallis test comparing network stability (average degree and network connectivity) of networks under No camping and Camping after 50% nodes were randomly removed. For each network, node properties were computed by bootstrapping 100,000 times.

Method	Comparison	Average degree	Natural connectivity	
Kolmogorov–Smirnov test	No camping B vs.
Camping B	0.2021****	0.0987****	
No camping F vs.
Camping F	0.9043****	0.8718****	
No camping B vs.
No camping F	0.9811****	1****	
Camping B vs.
Camping F	1****	1****	
Kruskal–Wallis test	No camping B vs.
Camping B	11.1137 vs. 11.6460****	42.3549 vs. 42.9296****	
No camping F vs.
Camping F	5.9364 vs. 2.9788****	18.3614 vs. 11.0958****	
No camping B vs.
No camping F	11.1137 vs. 5.9364****	42.3549 vs. 18.3614****	
Camping B vs.
Camping F	11.6460 vs. 2.9788****	42.9296 vs. 11.0958****	
Notes.

The values in top four cells represent difference for Kolmogorov–Smirnov test, which the maximum difference in the absolute cumulative distribution function; The values in bottom four cells represent mean value for Kruskal–Wallis test.

**** indicate statistical significance at p <0.0001.

B bacteria

F fungi

Figure 4 Network robustness analysis for microbial communities between the Camping and No camping.

Smaller decline at the same proportion indicates more stability within networks. B, bacteria; F, fungi.

Four types of changes in the microbial attributes were identified that could explain the change in network stability, which could help us to understand the impact of camping on microbial networks.

(1) More than 62.45–72.26% of high-degree OTUs and low-degree OTUs were specific to the no camping or camping, as indicated by Venn plots (Figs. 5A, 5B, 5E, 5F); this suggested that camping shifted the node identity of networks compared to no camping.

(2) Moreover, 17.86%–22.36% of OTUs were shared between camping and no camping, implying that camping shifted the high-degree OTUs and low-degree OTUs under the no camping from the states of high and low degree to the states of low and high degree, respectively (Figs. 5C, 5D, 5G, 5H). Furthermore, compared to no camping, camping decreased the node degree of Luteimonas, (OTU_287 and OTU_1365, from 77 and 77 to 1 and 0), Dyadobacter (OTU_1216 and OTU_3484, from 77 and 71 to 72 and 1), Ruminococcaceae UCG-005 (OTU_1880, OTU_399, OTU_4129 from 71,71,5 to1,1,2), Pseudanabaena PCC-7429 (OTU_3596, from 77 to 71), Family XIII AD3011 group (OTU_1230, OTU_594 from 3 and 77 to 2 and 0), Christensenellaceae R-7 group (OTU_2495, OTU_2549, OTU_4118, from 2, 3, 66 to 0, 0, 2), Prevotellaceae UCG-004 (OTU_2810, OTU_378, OTU_529, OTU_1946, OTU_3132 from 71, 77, 3, 4, 66 to 8, 4, 2, 1, 0), Olivibacter (OTU_446 from 77 to 72), OLB13 (OTU_2259 from 77 to 0), and Ruminococcus 1 (OTU_1411 from 66 to 0). Moreover, camping increased the node degree of Aquamicrobium (OTU_262, from 2 to 77), Limnothrix (OTU_99 from 0 to 72), Cellvibrio (OTU_219 0 to 72), FukuN18 freshwater group (OTU_392 from 0 to 77), OLB13 (OTU_446 from 1 to 71), Rudaea (OTU_2905 from 0 to 1), and RB41 (OTU_53 from 66 to 72), as well as some potential dung-derived phylotypes, including Ruminococcaceae UCG-005 (OTU_4330, OTU_1005, OTU_836 from 3, 1, 0 to 72, 4, 2), Christensenellaceae R-7 group (OTU_490 from 0 to 1), Prevotellaceae UCG-004 (OTU_2172, OTU_2408 from 2, 1 to 3, 10), and Ruminococcus 1 (OTU_449, OTU_539 from 0,6 to 10,10) (Tables S4 and S5), partially reflecting that some dung-derived phylotypes enhanced the network interaction.

(3) For the bacterial network, camping decreased the total relative abundance of high-degree OTUs (6.54% under camping vs. 7.27% under no camping) by 10.02% (Fig. 5A). Camping significantly decreased the niche overlap of high-degree OTUs and specific high-degree OTUs (Wilcoxon test, p = 6.5e−13, 7.0e−04), and significantly increased the relative abundance of high-degree generalist species and specific high-degree generalist species (Wilcoxon test, p = 1.1e−04, 1.1e−04) (Figs. S87A, S87C, S87E, S87H). Camping decreased the total relative abundance of low-degree OTUs (6.09% under camping vs. 6.28% under no camping) by 2.95% (Fig. 5B). Camping significantly increased the niche overlap of low-degree OTUs and specific low-degree OTUs (Wilcoxon test, p = 8.4e−4, 0.027) and decreased the niche breadth of low-degree OTUs and specific low-degree OTUs (Wilcoxon test, p = 1.1e−10, 8.2e−06) (Figs. S88A–S88D). However, camping increased the total relative abundance of the shared OTUs that shifted the states of high degree under no camping (1.87%) to the states of low degree under camping (2.08%) by 11.26% (Fig. 5C). Camping significantly decreased the niche breadth of these share degree OTUs (Wilcoxon test, p = 3.0e−05) (Fig. S89B). Camping decreased the total relative abundance of the shared OTUs that shifted the states of low degree under no camping (2.92%) to the states of high degree under camping (2.37%) by 18.84% (Fig. 5D). Camping significantly decreased the niche breadth of these share degree OTUs (Wilcoxon test, p = 1.9e−03) (Fig. S89D). For the fungal network, the camping decreased the total relative abundance of high-degree OTUs (1.24% under camping vs 9.85% under no camping) by 87.43% (Fig. 5E). Camping significantly increased the niche overlap of high-degree OTUs and specific high-degree OTUs (Wilcoxon test, p = 0.032, 0.011), significantly decreased the niche width of high-degree-OTUs and specific high-degree OTUs (Wilcoxon test, p = 2.8e−08, 1.9e−05), and significantly decreased the relative abundance of no significant species in high-degree OTUs and specific high-degree OTUs (Wilcoxon test, p = 0.0048, 0.0018) (Figs. S90A–S90D, S90E, S90H). Camping decreased the total relative abundance of low-degree OTUs (0.69% under campingvs. 1.49% under no camping) by 53.40% (Fig. 5F). Camping significantly increased the niche overlap and niche width of low-degree OTUs (Wilcoxon test, p = 0.018, 0.0002) and specific high-degree OTUs (Wilcoxon test, p = 0.19, 0.0005), and decreased the relative abundance of no significant species in low-degree OTUs and specific low-degree OTUs (Wilcoxon test, p = 0.045, 0.091) (Figs. S91A–S91D, S91F, S91I). However, camping decreased the total relative abundance of the shared OTUs that shifted the states of high degree under no camping (0.40%) to the states of low degree under camping (0.18%) by 54.95% (Fig. 5G). Camping increased the niche overlap and decreased the niche width of these shared OTUs (Wilcoxon test, p = 0.9 and p = 0.0056) (Figs. S92A, S91B). Camping increased the total relative abundance of the shared OTUs that shifted the states of low degree under no camping (0.16%) to the states of high degree under camping (1.03%) by 535.56% (Fig. 5H). Camping increased the niche overlap and decreased the niche width of these shared OTUs (Wilcoxon test, p = 0.041 and p = 9.0e−04) (Figs. S92C, S92D).

(4) For the bacterial communities, compared to no camping, camping significantly changed the relative abundance of functions of the low-degree OTUs, high-degree OTUs, specific low-degree OTUs and specific high-degree OTUs (Welch’s t test with two sides p = 7.73e−5–0.048, Figs.S93A–S93E). Moreover, compared with no camping, camping significantly suppressed the relative abundance of most C-, N-, and P-cycle functions and antibiotic resistance genes of the low-degree OTUs and specific low-degree OTUs, but improved those of high-degree OTUs and specific high-degree OTUs (Welch’s t test with two sides p = 2.28e−7–0.050, Figs. S94–S99). Camping significantly suppressed the relative abundance of most C-, N-, and P-cycle functions and antibiotic resistance gene of the shared OTUs that shifted the states of high degree under no camping to the states of low degree under camping; however, camping significantly improved the relative abundance of most C- and P-cycle functions and antibiotic resistance genes of the shared OTUs that shifted the states of low degree no camping under to the states of high degree under camping (Welch’s t test with two sides p = 9.84e−8–0.050, Figs. S100–S101). The relative abundance of C-, N-, and P-cycle functions and antibiotic resistance genes of high-degree OTUs and specific high-degree OTUs were significantly higher than those of low-degree OTUs and specific low-degree OTUs under camping, whereas the opposite trend was found under no camping (Welch’s t test with two sides p = 7.72e−7–0.049, Figs. S102–S110). For the fungal communities, compared to no camping, camping significantly decreased the relative abundance of Microfungus (Welch’s t test with two sides p = 0.047, 0.019, Figs. S111–S112A, S112D) of the low-degree OTUs and specific low-degree OTUs, improved the relative abundance of Undefined Saprotroph of the low-degree OTUs (Welch’s t test with two sides p = 1.43e−3, Fig. S111B), but decreased the relative abundance of Undefined Saprotroph of the specific low-degree OTUs (Welch’s t test with two sides p = 7.34e−3, Fig. S112E ). Furthermore, compared to no camping, camping significantly decreased the relative abundance of Saprotroph (Welch’s t test with two sides p = 3.0e−03, 0.012, Figs. S111–S112C, S112F), but improved the relative abundance of Symbiotroph (Welch’s t test with two sides p = 0.022, 0.016, Figs. S111–S112C, S112F) of both low-degree OTUs and specific low-degree OTUs, and declined the relative abundance in the white rot of both high-degree OTUs and specific high-degree OTUs (Welch’s t test with two sides p < 1.0e−15, < 1.0e−15, Figs. S111–S112G, S112H). Camping did not significantly change the relative abundance of functions of the shared OTUs that shifted the states of high degree under no camping to the states of low degree under camping (Welch’s t test with two sides p > 0.05); however, camping significantly suppressed the relative abundance of facultative yeast and yeast of the shared OTUs that shifted the states of low degree no camping under to the states of high degree under camping (Welch’s t test with two sides p = 1.0e−15–0.048, Figs. S111–S112I).

Figure 5 (A–H) Venn diagram showing the number of specific and shared nodes between different degrees and different groups (camping and no camping).

Niche breadth, niche overlap, specialist/generalist species and assembly mechanism of microbial communities under camping and no camping

Wilcoxon rank sum test revealed that the bacterial niche breadth was 1.2–1.3-fold those of fungi (p = 2.22e−16–1.3e−16), regardless of the camping and no camping. Compared to no camping, camping decreased the niche breadth index of bacteria and fungi by 3% (p = 0.054) and 14% (p = 1.9e−05), respectively (Fig. 6A). The Wilcoxon rank sum test revealed that the fungi niche overlap index was 1.2–1.4-fold that of bacteria (p < 2.22e−16), regardless of the camping and no camping. Compared to no camping, camping increased the niche overlap of bacteria and fungi by 2% (p < 2.22e−16) and 13% (p < 2.22e−16), respectively (Fig. 6B). T test suggested that the percentage of bacterial generalists was 4.1–4.7-fold that of fungi (p = 0.0022, 0.0016), while that of fungal specialists was 5.8-fold that of bacteria (p = 0.015, 0.0023), irrespective of the presence or absence of camping (Fig. 6C). Compared to no camping, camping increased the percentage of bacterial generalists and decreased the percentage of bacterial specialists by 3% (t test, p = 0.583) and 1% (t test, p = 0.953), respectively. Unexpectedly, camping decreased the percentage of fungal generalists and the percentage of fungal specialists by 10% (t test, p = 0.719) and 1% (t test, p = 0.953), respectively.

Figure 6 Differences in niche width (A), niche overlap (B), specialist/generalist species (C), assembly mechanism (D, E), and the node degree distribution with relative abundance ranks (F) of bacterial and fungal communities between camping and no camping.

Understanding how management and evolution shape community assembly mainly involves two opposing views: the niche and dispersal hypothesis. To identify the first-order drivers that drive community assembly, the dispersal–niche continuum index (DNCI) was used to quantify the relative importance of niche or dispersal process (Vilmi et al., 2020). The result showed that the E values from the niche-controlled model and niche—and dispersal-controlled models were lower than that from the dispersal-controlled model (Figs. 6D–6E), indicating that niche restriction was the main driving force of bacterial and fungal community assembly. The niche width linearly decreased the degree of networks (p = 0.0015 and p = 0.3652 for camping and no camping, respectively, Fig. 7A) and niche overlap increased the changes in the degree of networks (p < 2.2e−16 and p < 2.2e−16 for both camping and no camping, respectively, Fig. 7B). Furthermore, the Bayesian structural equation model suggested that camping exerted a positive effect on the network degree by compressing the niche width (Estimate = −0.02 to −0.10, −0.14 to −0.75, Fig. 7C), and camping reduced the change in the network degree by reducing the niche overlap (Estimate = −0.04 to −0.05, 0.26 to 0.29, Fig. 7D).

Figure 7 Linear regression model showing the effect of niche width (A) and niche overlap (B) on the network property (degree and difference between degrees (Δ degree)).

Bayesian structural equation model showing how camping alter the network property via niche width (C) and niche overlap (D).

Discussion

Herbivore camping changes the structure of subsoil microbial communities with functional implications

The present study was conducted to investigate the effects of herbivore camping on the taxonomy, function, and network of soil microbial communities. Earlier research had showed that herbivore camping affects the physical, and chemical properties (Jiang et al., 1999; Zhang et al., 1999b; Zhang, Jiang & Ren, 2001; Niu et al., 2009). However, no previous study has reported the subsoil bacterial and fungal communities in the context of herbivore camping. Previous studies have showed that camping changes the substance transformation rates (mineralized C, respiration, arginine ammonification), enzyme activities, and microbial biomass C and N, indicating altered microbial activity (Haynes & Williams, 1999; Iyyemperumal, Israel & Shi, 2007). These findings have improved the understanding of the impact of camping in the past and shed light on the impact of camping on microbial activity. However, we focused on bacterial and fungal community attributes (e.g., composition, functions, taxonomic and functional diversities, and network); therefore, in this sense, our study firstly demonstrated that camping significantly enriched and suppressed some bacterial and fungal abundances.

Camping-induced alteration in bacterial and fungal abundances could result in microbial community function fluctuations, and has important implications for soil carbon and nutrient cycling, and soil and plant health. Regarding the bacterial genus that were changed by camping, Luteimonas, Dyadobacter, Ruminococcaceae UCG-005, Aquamicrobium, Cohnella, Limnothrix, Christensenellaceae R-7 group, Prevotellaceae UCG-004, Olivibacter, Rummeliibacillus, OLB13, Ruminococcus 1, LD29, Rudaea, RB41, Cellvibrio, and Clostridium sensu stricto 18 have been found to be involved in the degradation of various carbohydrates, as well as CO2 fixation (Table S5) (Yannick & David, 2013; Zhang et al., 2017; Diana et al., 2020). The genera Chelatococcus, Luteimonas, Dyadobacter, Aquamicrobium, Cohnella, Olivibacter, OLB13, Rudaea, RB41, and Cellvibrio are involved in denitrification (Yannick & David, 2013), nitrification (Zhang et al., 2019; Mekdimu et al., 2021), ammonia oxidation (Su et al., 2021), nitrogen fixation (Xiao et al., 2019), respiratory ammonification, and nitrogen assimilation (Meier et al., 2021). Luteimonas, Dyadobacter, Olivibacter, Rummeliibacillus, Cellvibrio are reported to promote plant growth (Saurabh et al., 2018; Akyol, Ince & Ince, 2019; Cristóbal et al., 2022) by increasing nutrient (Kohout, 2019) and/or disease suppression (Mohamed et al., 2017; Fu et al., 2017; Kristina et al., 2020) (Table S5). As Rudaea genus is pathogenic bacteria (Li et al., 2021b), the higher abundance of the genera (Chelatococcus, Luteimonas, Dyadobacter, Ruminococcaceae UCG-005, Aquamicrobium, Cohnella, Limnothrix, Pseudanabaena PCC-7429, Christensenellaceae R-7 group, Prevotellaceae UCG-004, Olivibacter, OLB13, and Rummeliibacillus) in camping than suggested that camping could be beneficial to soil carbon and nitrogen cycling and health. Regarding the fungal genera, Podospora, Symbiotaphrina, and Rhizomucor are degraders of various plant materials (Table S5) (He et al., 2019), and contribute to element cycling. Podospora is known to antagonise soil-borne diseases (Tao, Hu & Chu, 2020) and enhance root growth (Yim et al., 2017), and is therefore most abundant in healthy soils (Xu et al., 2012). The higher abundance of this genus in camping compared to no camping suggested that camping could improve the soil health. However, Rhexocercosporidium (e.g., R. panacis) is a phytopathogenic fungus commonly found in soils (Douterelo et al., 2016) and can cause ginseng rusty root rot, rusted roots of ginseng (Table S5). The higher abundance of this genus in camping compared to no camping suggested that camping negatively impact the health of some plants. Significantly, Rhizomucor genus is a human opportunistic pathogen that frequently causes fatal mycotic diseases (Gyöngyi et al., 2004). Studies suggested that fungi from the Candida genus are critical human and animal pathogens and are drug-resistant (Karpinski et al., 2021) (Table S5). As the pathogenic fungal genera are reduced, the camping area becomes safer for human and animals. In conclusion, camping-induced alteration in bacterial and fungal abundances may be beneficial to plant, human and animal health, representing a novel finding of this study.

Interestingly, camping animals may disseminate some microorganisms through feces. Semenov et al. (2021) found that some genera that were transmitted from dung was also predominant among metabolically active phylotypes in dung-treated soil. In this study, Family XIII AD3011 group, Prevotellaceae UCG-004, Christensenellaceae R-7 group, Ruminococcus 1, Ruminococcaceae UCG-005, Prevotellaceae UCG 004, LD29, Olivibacter, and Symbiotaphrina were also detected in the rumen, gut and/or dung (Table S5), suggesting that the subsoil microbial community could be affected by microorganisms from the dung of camping herbivores. More interestingly, a recent study found that most microorganisms from dung did not survive in soil beyond a few months (Semenov et al., 2021), suggesting that the influence of camping on the subsoil microbial communities could be temporary, although this needs further study.

Importantly, camping may have a negative effect on antibiotic resistance genes. The Cellvibrio genera carry the tetracycline resistance gene (Table S5),and depletion of Cellvibrio could contribute to the decrease in the relative abundance of tetracycline resistance gene (tetB). Tetracyclines are the best-selling veterinary antibiotic, and are found at the highest content in animal manures (Yue et al., 2021). Our results showed the abundance of tetracycline resistance gene (tetB) was significantly depleted by camping, suggesting that camping could be a potential method for eliminating of fecal antibiotic resistance genes. This finding has not been reported, previously and there remains a lack of research on this topic.

It is worth mentioning that camping changed the niche space of microorganisms. The lower abundance of the aerobic genus LD29 (Zhao et al., 2021) and higher abundance of the anaerobic genus OLB13 and Rummeliibacillus under camping compared to under no camping suggested a hypoxic environment under camping. This may reflect previous findings that camping and trampling reduced soil air permeability and caused soil hypoxia (Hou et al., 2004).

Herbivore camping affects the network of subsoil bacterial and fungal communities

In this study, camping increased the bacterial and fungal network complexity, particularly the number of positive edges. The results are consistent with the findings demonstrating the impact of manure on the microbial network (Ye et al., 2021). Four potential mechanisms may explain this phenomenon: (1) The stress-gradient hypothesis and our previous study on microbial interactions showed that stressed habitats facilitates (positive links) higher frequency than competition (negative links) (Ding & Wang, 2021). As discussed above, camping and trampling could reduce soil air permeability and cause soil hypoxia (Hou et al., 2004). Additionally, camping and dung and urine input resulted in high soil ammonia concentrations (up to 377 mg/kg) (Zhang, 2002) which may be toxic to microbes. Both hypoxia and high ammonia can stress the soil microbes, leading to more positive links. (2) Many studies (Kong et al., 2019; Torres, Yu & Kurtural, 2021) have shown that microbial inoculation could increase the positive interaction of microbial network. Microorganisms originating from the addition of manure could also enhance the association of microorganisms (Yang et al., 2022). Indeed, some potential dung-derived phylotypes, including Ruminococcaceae UCG-005, Christensenellaceae R-7 group, Prevotellaceae UCG-004, Ruminococcus 1 (Table S5) elevated their node degree in networks under camping compared to no camping. (c) Microorganisms that cooperate with the original soil microorganisms were more likely to successfully colonize and proliferate in the soils. This scenario could also enhance positive microbial interactions. (4) Nutrient input from feces and urine led some microorganisms to proliferate, which was reflected by the increase in microbial carbon and nitrogen (88% and 84%, respectively) following the manure addition (Liu et al., 2020). These synergistic increases could also lead to more positive links. Besides, some low abundance bacterial and fungal taxa had the highest degree of the network, suggesting that low rather than high -abundant bacteria and fungi were keystone taxa that affect the stability of the network, supporting previous findings (Pan et al., 2021). To our knowledge, this is the first study to show that herbivore camping could affect the stability of the network by shifting the properties of high- and low- degree taxa. Furthermore, for the first time, the evidence indicated by our Bayesian structural equation models provides useful knowledge for managing camping soil microorganisms in pasture and for insights into the impact of wild camping behavior on soil that have been largely overlooked.

Conclusion

Our results first suggested that camping significantly changed the relative abundance of 21 bacterial phylotypes and five fungal phylotypes, with implications for soil carbon and nutrient cycling, and soil and plant health, but did not change taxonomic and functional diversity. Compared to no camping, camping increased the stability of the bacterial network, whereas, camping decreased the stability of the fungal network. Camping exerted a positive effect on the network by compressing the niche width, and reduced the change in the network by reducing the niche overlap. However, the effects of this change on the soil viruses, protozoa, and the plants that are established later remain unknown. This study provides a stepping-stone insight into the effect of herbivore camping on soil microbial communities.

Supplemental Information

Supplemental Information 1 Supplementary Materials

Click here for additional data file.

Additional Information and Declarations

Competing Interests

Author Contributions

DNA Deposition

Data Availability

The authors declare there are no competing interests.

Puchang Wang conceived and designed the experiments, performed the experiments, authored or reviewed drafts of the article, funding acquisition, and approved the final draft.

Leilei Ding conceived and designed the experiments, performed the experiments, analyzed the data, prepared figures and/or tables, authored or reviewed drafts of the article, funding acquisition, and approved the final draft.

Fuxiang Li performed the experiments, prepared figures and/or tables, and approved the final draft.

Jiafa Liao performed the experiments, prepared figures and/or tables, and approved the final draft.

Mengya Wang performed the experiments, prepared figures and/or tables, authored or reviewed drafts of the article, and approved the final draft.

The following information was supplied regarding the deposition of DNA sequences:

The raw data is available at NCBI: SAMN29636188, SAMN29636189, SAMN29636190, SAMN29636191, SAMN29636192, SAMN29636193 for 16S and SAMN29636240, SAMN29636241, SAMN29636242, SAMN29636243, SAMN29636244, SAMN29636245 for ITS.

It is also available at figshare: Ding, Leilei (2022): rawdata. figshare. Dataset. https://doi.org/10.6084/m9.figshare.20277978.v1.

The following information was supplied regarding data availability:

The supplementary materials, data and codes are available at Github: https://github.com/dlltargeting/camping-shifts-microbiome, Leilei Ding. (2022). camping-shifts-microbiome. Zenodo. https://doi.org/10.5281/zenodo.7192408.

The raw data is available at figshare: Ding, Leilei (2022): rawdata. figshare. Dataset. https://doi.org/10.6084/m9.figshare.20277978.v1.

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
