# Peer review of "Herbivore camping reshapes the taxonomy, function and network of pasture soil microbial communities"

_PeerJ, doi:10.7717/peerj.14314_

## Round 0.1 · original submission · Major Revisions

Dear Dr. Ding,

As per the recommendations and suggestions provided by our expert reviewers, the manuscript has to go under major revision. However, all or reviewers appreciated the concept and the area of your research. It seems that the language of the manuscript has more scope to be improved therefore, I advise you to take help from professional language editors.

Please incorporate or justify (if you disagree with the reviewer's opinion) the corrections to improve the manuscript and resubmit asap.
regards and good luck.

·

Basic reporting

This research article is good for publication; however, some of the points need to be addressed before the publication of this article. All the points have been mentioned below.

Experimental design

It is well designed. See my comments in additional comments.

Validity of the findings

The findings are good. See my comments in additional comments.

Additional comments

This research article is good for publication; however, some of the points need to be addressed before the publication of this article. All the points have been mentioned below:

1. The abstract lacks a conclusion section. I suggest including the conclusion part at the end of the abstract.
2. The discussion is too lengthy. The section should be deducted and compressed. Only important findings should be highlighted and their significance should be briefly discussed. I suggest removing the subheading from the discussion section.
3. The English language requires revisions.
4. Many of the references are not as per the format of the journal. Update accordingly.
5. I appreciate the authors for their efforts in this study. The figures used in this study are very good.

Rest is ok.

Reviewer 2 ·

Basic reporting

Current manuscript studied the effect of 15 days camping of sheep on soil microbial community. Author collected the soil (10-20 cm layer) from the camping site (three samples were collected and pooled together) on 16th day of camping. For non-camping site, author used nearby site where topographies, plants, and soil type are nearly identical and collected three samples and pool them together.
DNA was extracted from soil and analyzed for bacterial and fungal community using 16S ribosomal RNA gene (for bacteria) and internal transcribed spacer (ITS) 2 region of ribosomal gene (for fungi) sequencing.
Author concluded that camping increased the relative abundance of 21 bacterial phylotypes and five fungal phylotypes however it is very difficult for reviewer to find out which bacteria or fungal phylum is increased.
Overall manuscript has good theme to determine the effect of herbivore animal comping on soil microbial community but there are many loopholes in the manuscript which need to be fixed which as follows:
1. Manuscript need an extensive re-writing for grammar, sentence punctuation so that author can read and understand the message from the current work.
2. Author stated at line 21-22, “herbivore camping on soil physicochemical properties have been studied. whether the change soil microbial communities remain unknown, especially below the surface” and at line 35-36 “This study provides a first insight into the of animal camping on soil microbial communities for grassland” however there are many studies which found that animal camping affect the soil chemical and microbial community (soil samples taken for those studies were as deep as 0-20 cms)

Few of the studies are as follows while other studies author already mentioned in the manuscript. Therefore, either authors either need to re-write those sentences and rebuttal how there work is 1st in determining the effect camping on soil microbes and how it is different than others

• Vargas et al. 2015. Microbial quality of soil from the Pampa biome in response to different grazing pressures. Genet Mol Biol. 38(2): 205–212
• Lyyemperumal et al. 2007. Soil microbial biomass, activity and potential nitrogen mineralization in a pasture: Impact of stock camping activity. Soil Biology and Biochemistry 39(1):149-157
• QI, et al. 2011. Effects of livestock grazing intensity on soil biota in a semiarid steppe of Inner Mongolia. Plant and Soil 340(1):117-126
• Haynes and Williams. 1999. Influence of stock camping behaviour on the soil microbiological and biochemical properties of grazed pastoral soils. Biology and Fertility of Soils volume 28, pages253–258.

3. Line 132-134: “For surface soil (0-10 cm), the organic carbon content was 61.05 g/kg, total nitrogen content 3.88 g/kg, total phosphorus content 1.01 g/kg, alkali hydrolyzed nitrogen content 354.04 mg/kg, and available phosphorus content 6.43 mg/kg”:
How author measured organic carbon, nitrogen, phosphorus and alkali hydrolyzed nitrogen content is not clear (no methodology is mentioned). If author used the data for there chemical composition from other study, then author need to cite the reference (keeping in mind that, if study has not been done recently, then chemical composition on solid may be changed from last time it was analyzed).

4. Author mentioned chemical composition for soil layer 0-10 cm and 10-20 cm, while for bacterial analysis from camping area, soil was taken from 10-20 cm layer (line 144). If it is true, author need to explain why they did not take the upper layer where more fecal matter come in contact with soil?

5. Author did not motion any aseptically technique in collecting the soil sample as contamination may change the bacterial composition of sample.

6. Author did not mention if soil sample were kept cold (liquid nitrogen, dry ice) until DNA is extracted as longer storage may also change bacterial composition.

7. Author mentioned that, they collected soil sample from camping site (3x and pooled together) and non-camping site (3x and pooled together) (Line 145-146), with this practice, authors had one sample for each treatment (camping vs non-camping). How it is possible to perform ANOVA or T test with one sample?

8. Author mentioned several suppletory figures and tables in the manuscripts (line 242-252) but review could not find any of these supplementary materials.

Experimental design

Experimental design is very vague. Did not clearly mentioned how many soil samples were collected. For a statistical analysis we should have sufficient sample numbers (depending up on confidence level), which were missing here.

Title states “Herbivore camping reshapes the taxonomy, functions and network of pasture soil microbial communities” while there is no functional analysis is done in the manuscript. Author did functional analysis based relative bacterial composite in soil (page 242-264). Which is again depends upon number of samples analysis in the study

Validity of the findings

Submitted figures are good but several supplementary figures and tables which are mentioned in the text are missing

·

Basic reporting

First of all authors are appreciated for the extensive investigation peformed on the topic.

BASIC REPORTING

The language and grammar required major revision in the whole manuscript.

The structure and presentation of the manuscript is not systematic, which may kindly be improved.
Certain corrections are mentioned in the manuscript which may kindly be included.

Introduction
Most humbly it is brought to your kind notice that the introduction was found to be extensive, vague and not following the standard structure. The following suggestions may be followed for rewriting the introduction:
• first the topic is introduced with standard but brief chronological review up to recent times,
• Based on introduction, then the problem/ lacuna in research is recognized and hypothesis is put forward.
• Then based on hypothesis, objectives of the investigation are given.
It is suggested to kindly avoid reproducing data from the cited research and keeping statements brief with correct language, stating only the essence of their findings.
Literature well referenced & relevant, however needs structured arrangement.


Figures are relevant, high quality, well labelled & described properly .

Raw data was supplied

Experimental design

EXPERIMENTAL DESIGN
The presented research is original primary research within Scope of the journal.

Research question were not well defined in the introduction or material and methods however are relevant & meaningful.

It is stated how the research fills an identified knowledge gap.

Rigorous investigation was performed to a high technical & ethical standard.

Comments for Material Methods-
Most humbly it is brought to your kind notice that the material method was found to be written in a disorganized manner.
Kindly mention each test separately by including a statement to emphasize its relevance in the study.
Where ever needed a protocol or formula may kindly be mentioned with citation of proper original reference. If a test or protocol had been earlier used by authors or any other worker then mention of such citation may kindly be restricted to only discussion and not in the material and methods.
The standard protocols or manufacturers protocols in kits should not be given citations from the previous publications of authors.
The material and method may kindly mention a statement about following the guidelines of ethical treatment to animals.

Validity of the findings

VALIDITY OF THE FINDINGS
The research will have good impact in concerned field and it is a unique novel work.

Following observation regarding discussion are mentioned.
Most humbly it is brought to your kind notice that the discussion was found to be extensive, vague and not following the standard structure. The following suggestions may be followed for rewriting the discussion:
• First the topic is introduced by mentioning the hypothesis and objectives in brief with standard but brief chronological review up to recent times. For example “the presented study was conducted to investigate ………”. Followed by brief review “earlier research had showed …………”
• Results may kindly be discussed one by one without repeating the already mentioned findings again.
• The discussion of each finding may kindly start with mentioning the reason why the test was conducted or by stating the relevance of the test conducted for the presented study. If required a citation may be given.
• The discussion of each finding may kindly be clear cut and direct in comparison. Each finding discussion should end in a conclusive statement like- “the findings of the test clearly indicate that ……………” . At this point the novelty of findings may be mentioned against the cited research to show the impact of presented research.
• Discussion may end with remarks briefly mentioning the finding in logical sequence to draw a meaningful conclusion.
It is suggested to kindly avoid reproducing data from the cited research as far as possible until very necessary.
It is suggested to kindly keep review statements brief with correct language, stating only the essence of their findings which are comparable to the presented study.
Since results are given separately therefore, kindly do not mention the tables and figures in discussion.

---

## Round 0.2 · accepted · Accept

It is my pleasure to inform you that as per the recommendation of our expert reviewers, the manuscript "Herbivore camping reshapes the taxonomy, function and network of pasture soil microbial communities" - has been Accepted for publication in PeerJ.

This is an editorial acceptance and you will be contacted with a list of further tasks before publication. So, I request you to be available for a few days to make the necessary things asap.

Regards and good luck with your future submissions.

·

Basic reporting

Good.

Experimental design

Good.

Validity of the findings

Good.

Additional comments

None.

·

Basic reporting

All recommended corrections included.

Experimental design

Rigorous investigation performed to a high technical & ethical standard. All recommended corrections included

Validity of the findings

Findings are well defined and described with proper scientific evidence.

Additional comments

Research contributes important information to the field of study.